# Covariant spatio-temporal receptive fields for spiking neural networks

J. E. Pedersen ✉, J. Conradt & T. Lindeberg

Biological nervous systems constitute important sources of inspiration towards computers that are faster, cheaper, and more energy efficient. Neuromorphic disciplines view the brain as a coevolved system, simultaneously optimizing the hardware and the algorithms running on it. There are clear efficiency gains when bringing the computations into a physical substrate, but we presently lack theories to guide efficient implementations. Here, we present a principled computational model for neuromorphic systems in terms of spatio-temporal receptive fields, based on affine Gaussian kernels over space and leaky-integrator and leaky integrate-and-fire models over time. Our theory is provably covariant to spatial affine and temporal scaling transformations, with close similarities to visual processing in mammalian brains. We use these spatio-temporal receptive fields as a prior in an event-based vision task, and show that this improves the training of spiking networks, which is otherwise known to be problematic for event-based vision. This work combines efforts within scale-space theory and computational neuroscience to identify theoretically well-founded ways to process spatio-temporal signals in neuromorphic systems. Our contributions are immediately relevant for signal processing and event-based vision, and can be extended to other processing tasks over space and time, such as memory and control.

Brain-inspired neuromorphic algorithms and hardware are heralded as a possible successor to current computational devices and algorithms[1,2]. Mathematical models that approximate biological neuron dynamics are known for their computational expressivity[3], and neuromorphic chips have shown to operate faster and with much less energy than computers based on the von Neumann architecture[1]. These "biomorphic" neural systems are decentralized and sparsely connected, which renders most sequential algorithms designed for present computers intractable. A few classical algorithmic problems have successfully been recast to neuromorphic hardware, such as MAXCUT[4] and signal demodulation[5], but no neuromorphic algorithm or application has, yet, been able to perform better than digital deep learning techniques[2]. Before we can expect to exploit the possible advantages of neuromorphic systems, we must first discover a theory that accurately describes these systems with a similar precision as we have seen for digital computations[6]. Without such a theoretical framework, neuromorphic systems will routinely be outperformed by dense and digital deep learning models[7].

Scale-space theory studies signals across scales[8] and has been widely applied in computer vision[9], and, recently, deep learning[10–12]. Lindeberg presented a computational theory for visual receptive fields that leverages symmetry properties over space and time[13], which is appealing from two perspectives: it provides a normative view on visual processing that is remarkably close to the stages of visual processing in higher mammals[14], and it provably captures natural image transformations over space and time[15]. Scale-space theory is closely tied to covariance (or equivariance), which is a desirable property because it optimally encodes and generalizes transformations in the feature space. Covariance has gained popularity in the machine learning literature[16] and examples are abundant in classical computer vision, where scale invariance and covariance for image and video data under varying image transformations have been proposed, including

Computational Science and Technology, KTH Royal Institute of Technology, Stockholm, Sweden. ✉e-mail: jeped@kth.se

objects of different size in purely spatial images[17–20], objects varying in projected image size due to depth variations over time[21], or objects and events under simultaneous spatial and temporal scaling variations[22–24]. However, there have not been any previously reported treatments of scale-space for event-based vision or joint spatio-temporal scale channels for deep learning on video data.

This work sets out to improve our understanding of spatio-temporal computation for event-based vision from first principles, with an emphasis on theoretically optimal encodings of objects under spatial and temporal transformations. We study the idealized computation of spatio-temporal signals through the lens of scale-space theory and establish covariance properties under spatial and temporal image transformations for biologically inspired neuron models (leaky integrator and leaky integrate-and-fire). Applying our findings to an object tracking task based on sparse, simulated event-based stimuli, we find that our approach benefits from the idealized covariance properties and significantly outperforms naïve deep learning approaches.

Our main contribution is a computational model that provides falsifiable predictions for event-driven spatio-temporal computation. We additionally provide experimental evidence that demonstrates the benefits of our approach in several event-based object tracking tasks. While we primarily focus on event-based vision and signal processing, we discuss more general applications within memory and closed-loop neuromorphic systems with relevance to the wider neuromorphic community.

Specifically, we contribute with (a) biophysically realizable neural primitives that are covariant to spatial affine transformations, Galilean transformations as well as temporal scaling transformations, (b) a normative model for event-driven spatio-temporal computation, and (c) an implementation of stateful biophysical models that perform better than stateless artificial neural networks in an event-based vision regression task.

Representing signals at varying scales was studied in modern science in the 70's[25,26] and formalized in the 80's and 90's[8,9]. Specifically, scale spaces were introduced by Witkin in 1983, who demonstrated how signals can be processed over a continuum of scales[27]. In 1984, Koenderink introduced the notion that images could be represented as functions over three coordinate variables $L : \mathbb{R}^2 \times \mathbb{R}_+ \to \mathbb{R}$, where a scale parameter represents the evolution of a diffusion process over the image domain[8]. Operating with diffusion processes, represented as Gaussian kernels, is particularly attractive because they parameterize scaling operations in the image plane as a linear operation[9]. In turn, this permits a concise and correct description of transformation in the signal plane, which has successfully been leveraged in algorithms for computer vision. For instance, in the early convolutional networks[21,28], but also in recent deep learning papers under the name equivariance with extension to other transformations than scaling under the guise of group theory. Specifically, they consider group actions under certain symmetric constraints, such as the circle $\mathbb{S}^1 \simeq SO(2)$[11,16,29]. Related to scale-space theory, Lindeberg introduced covariance properties for more general spatial transformations[13], and Howard et al.[30] revisited temporal reinforcement learning and identified temporal scale covariance in time-dependent deep networks as a fundamental design goal. Finally, we refer the reader to a large body of literature that studies general covariance properties for stochastic processes[31].

Mathematical models of the electrical properties in nerves date back to the beginning of the 20th century, when Luis Lapicque approximated tissue dynamics as a leaky capacitor[32]. The earliest studies of computational models appear in Lettvin et al.[33], who identified several visual operations performed on the image of a frog's brain, and in Hubel and Wiesel's[34] demonstration of clear tuning effects of visual receptive fields in cats, which allowed them to posit concrete hypotheses about the functional architecture of the visual cortex.

Many postulates have been made to formalize computations in neural circuits since then, but studying computation in individual circuits has only "been successful at explaining some (relatively small) circuits and certain hard-wired behaviors"[35, p. 1768]. We simply do not have a good understanding of the spatio-temporal computational principles under which biophysical neural systems operate[1,6,7].

The notion of scale-space theory relates directly to sensory processing in biology, where multiple distributed time-scales are known to play a critical role[33,36]. For instance, in time-invariant dynamics of neural memory representations[30] and the spatial components of visual receptive fields[13,14]. In addition[37], demonstrated that heterogeneous time constants metabolically efficient way to represent multiple time scales, particularly if they are tuned to the time scales in the tasks. More recently, scale-space theory has enhanced the performance of deep convolutional neural networks significantly[38] while enabling invariance to time-scaling operations[39]. Applying banks of Gaussian derivative fields, as arising from scale-space theory, has been demonstrated to express provably scale-covariant and scale-invariant deep networks[12].

Inspired by biological retinas, the first silicon retina was built in 1970[40] and independently by Mead and Mahowald in the late 1980s[41]. Event-based cameras have since then been studied and applied to multiple different tasks[7,42], ranging from low-level computer vision tasks such as feature detection and tracking[43,44] to more complex applications like object segmentation[45], neuromorphic control[46], and recognition[47].

In 2011, Folowsele et al.[48] applied receptive fields and spiking neural networks to perform visual object recognition. In 2014, Zhao et al.[49] introduced cortex-like Gabor filters in a convolutional architecture to classify event-based motion detection, closely followed by Orchard et al.[50]. In 2017, Lagorce et al.[51] formalized the notion of spatial and temporal features in spatio-temporal time surfaces, which they composed in hierarchies to capture higher-order patterns. This method was extended the following year to include a memory of past events using an exponentially decaying factor[52]. Schaefer et al.[53] used graph neural networks to parse subsampled, but asynchronous events in time windows. More recently, Nagaraj et al.[54] contributed a method based on sparse, spatial clustering that differentiates events belonging to objects moving at different speeds.

## Results

We establish spatio-temporal covariance for event-driven leaky integrator and leaky integrate-and-fire neuron models by extending the generalized Gaussian derivative model for spatio-temporal receptive fields[14,22]. To validate the theory, we later demonstrate by experiments that the spatio-temporal receptive fields, encoded with priors from the theory, provide a clear advantage compared to baseline models.

### Joint covariance under geometric image transformations

Image data are subject to natural image transformations, which, to first-order of approximation, can be modeled as a combination of three types of geometric image transformations:

1. spatial affine transformations $x' = A x$, where $A$ is an affine transformation matrix,
2. Galilean transformations $x' = x + u t$, where $u$ is a velocity vector and $t$ denotes time, and
3. temporal scaling transformations $t' = S_t t$, where $S_t$ is a temporal scaling factor.

Scale-space theory prescribes that signals subject to the transformations above can be represented in a covariant way by a scale-space representation $L$[14,15]. In addition, we can know the exact transformations between two scale-space representations $L$ and $L'$, provided that the parameters of the capturing mechanism are matched to the image transformation. That is, the image transformations

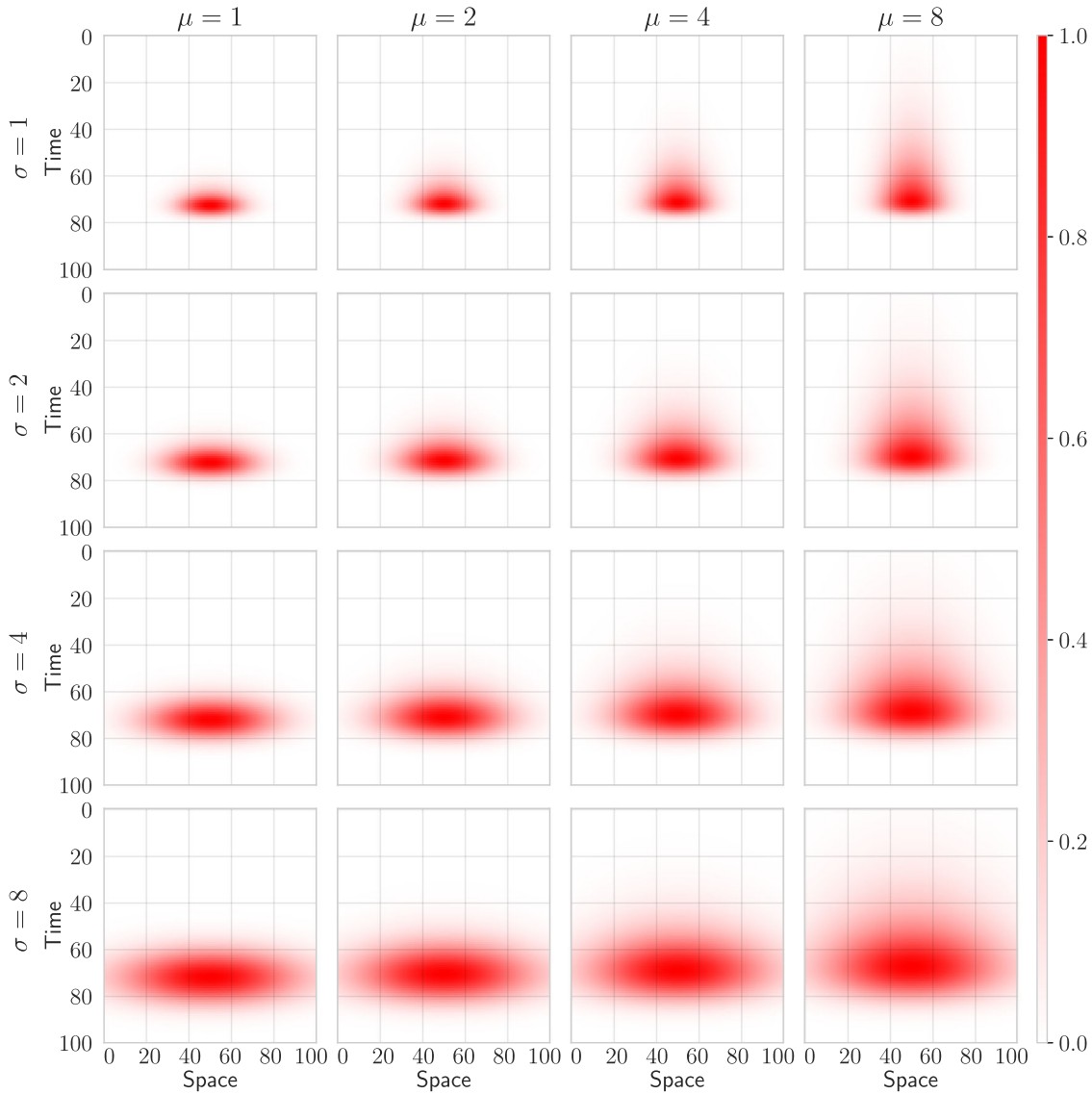

**Fig. 1 | Spatio-temporal receptive fields parameterized over 4 spatial scales ($\sigma$) and 4 temporal scales ($\mu$).** By increasing the size of the receptive field (rows), receptive fields become sensitive to larger spatial sizes. By increasing the temporal scale of the receptive fields (columns), the receptive fields become sensitive to larger temporal scales. The receptive fields in this figure have been normalized to [0, 1].

and scale-space smoothing operations commute, as detailed in the Methods Section Relationship under geometric image transformations and Supplementary Materials Section B.

Figure 1 illustrates 4 × 4 spatio-temporal receptive fields that are initialized to capture signals that have been scaled over space and time. The horizontal axes in the plots represent a 1-dimensional kernel that is scaled to match different spatial scales across the rows of the figure. The vertical axes represent the amount of temporal smoothing, which varies across the columns. By applying each receptive field to a given signal, we get the canonical scale-space representation at the given spatial and temporal scales.

To be sensitive to the spatial and temporal scales in any given data distribution, care must be taken to sample the parameter space accordingly. To explicitly handle variable spatial and temporal scales in the event-based video data, we filter the input data with a distribution of spatial receptive fields over a set of different spatial scales, that are in turn also expanded over 4 temporal scales, as detailed in the Methods Section on Model architecture and training.

The commutative relationship between scale-space smoothing operations and natural image transformations enables us to capture

fundamental spatio-temporal video transformations. This theoretical foundation permits us to model how visual information changes across both space and time (shown mathematically in Fig. 6 of the Methods Section). Models must accurately reflect the underlying data distribution through properly calibrated spatial and temporal parameters. When these parameters are misaligned, the model fails to capture essential transformations in the data, effectively becoming blind to important changes.

This alignment between spatial and temporal receptive fields and the underlying data is critical for accurate visual processing. Figure 2 illustrates this joint covariance property through a case study of spatial and temporal scaling operations. The experiment tracks a 2-dimensional square (panels a, b) as it scales over 75 time steps at three different velocities (panel a). By $t = 75$, these varying velocities result in squares of significantly different sizes (panel b). Panel (c) reveals how spatio-temporal receptive fields respond covariantly to these scaled squares. Remarkably, despite substantial differences in square sizes, the outputs remain similar across scales. Different temporal scale factors ($\mu$) produce functionally equivalent responses, regardless of whether scaling occurs in the input signal or evolves over

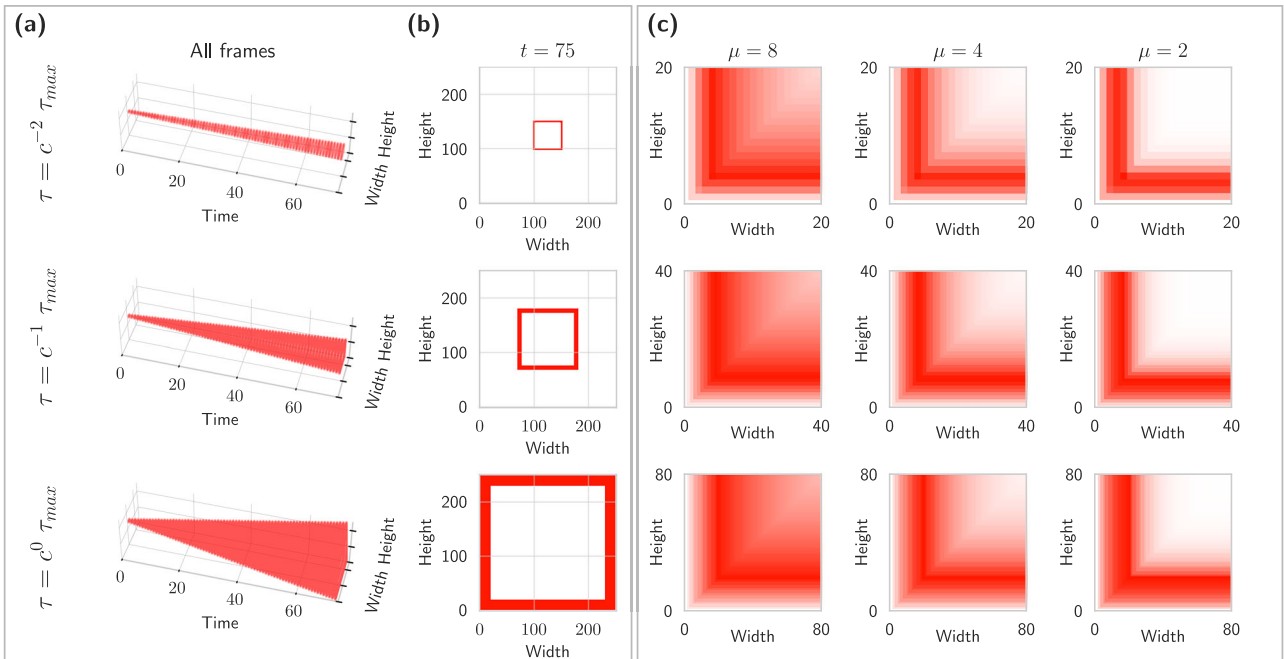

**Fig. 2 | Illustration of the combined spatial and temporal scale covariance properties of the underlying spatio-temporal receptive field representation.** When we scale three identical squares at different scale velocities $\tau$ over 75 time-steps in panel (**a**), according to Equation (16), we observe a large variance in the scale at the final timestep $t = 75$. Panel (**b**) shows the square signal as it appears at $t = 75$. Panel (**c**) shows the receptive field responses of the stimulus in (**a**) for different temporal scale parameters $\mu$ in panel. The responses appear similar, but occur at largely different spatial scales, because the spatial scale velocities $\tau$ are matched to the parameters of the geometric image transformations. In the ideal continuous case, they would be identical.

time. This scale-matching property significantly enhances the performance of computer vision algorithms that utilize spatio-temporal receptive fields, as it aligns naturally with geometric image transformations. The result is a more robust and theoretically grounded approach to video analysis.

### Temporal scale covariance for leaky integrators and leaky integrate-and-fire models

Under the image transformations above, we extend the notion of the temporal processing in the scale-space representations $L$ and $L'$ to leaky integrators and leaky integrate-and-fire models. When reduced to a purely temporal domain under a temporal scaling transformation of the form $t' = S_t\, t$, where $S_t > 0$ is the temporal scaling factor, a scale-covariant purely temporal scale-space representation should obey[55], Eq. (10) for $n = 0$[56], Eq. (23)

$$L(t;\, \tau) = L'(t';\, \tau') \tag{1}$$

where $\tau$ and $\tau' = S_t^2\, \tau$ are matching temporal scale parameters. A specific type of temporal scale channel representation $L$ for some signal $f$ can be written as ref. 57, Eq. (14)

$$L(t;\, \mu) = \int_0^\infty f(t - u)\, h_{exp}(\cdot;\, \mu)\, du, \tag{2}$$

where $\mu$ is a time constant and $h_{exp}(\xi;\, \mu) = \frac{1}{\mu}\, e^{-\xi/\mu}$. The leaky integrator can be understood as the integration of a decaying exponential function of some input signal $I$ [58], Eq. (1.11)

$$u(t;\, \mu) = \int_0^\infty \frac{1}{\mu}\, e^{-\xi/\mu}\, I(t - \xi)\, d\xi. \tag{3}$$

By reformulating $f$ and $h$, we observe that the two temporal representations in Equations (20) and (21) are covariant under temporal

scaling transformations. A full derivation is available in the Methods Section, Temporal scale covariance for leaky integrator models.

The leaky integrate-and-fire model expands the leaky integrator model with a Heaviside threshold, which causes the neuron to emit a spike and reset the temporally integrated value $u$ to 0. Assuming that the membrane reset can be expressed as a rapid, but linear, decay, we can express the leaky integrate-and-fire model as a series of kernels according to the Spike Response Model (SRM) described in the Methods Section Spike response model

$$z(t) = -\theta_{thr}\, e^{-(t - t_f)/\mu_r} + \int_0^\infty \kappa(s)\, I(t - s)\, ds, \tag{4}$$

where $\theta_{thr}$ is the threshold, $\mu_r$ is the reset time constant, and $\kappa$ is the linear membrane filter. Under mild assumptions, we show that the leaky integrate-and-fire model is also covariant to temporal scaling transformations.

In the Methods Section, Temporal scale covariance for leaky integrate-and-fire models, we provide the full exposition of how we can make the leaky integrate-and-fire model covariant under temporal scaling transformations.

A main corollary of these results is that, by replacing the previous temporal smoothing kernels in the spatio-temporal receptive fields in the spatio-temporal scale-space representations[14,15,22] by either leaky integrators or leaky integrate-and-fire models, we can construct joint covariant spatio-temporal receptive fields in spiking neural networks, which can in theory, exploit the symmetry properties of the underlying transformations that objects in event-based vision data are subject to.

### Initializing deep networks with idealized receptive fields

It is well known that spiking neural networks are comparatively harder to train than non-spiking neural networks for event-based vision, and

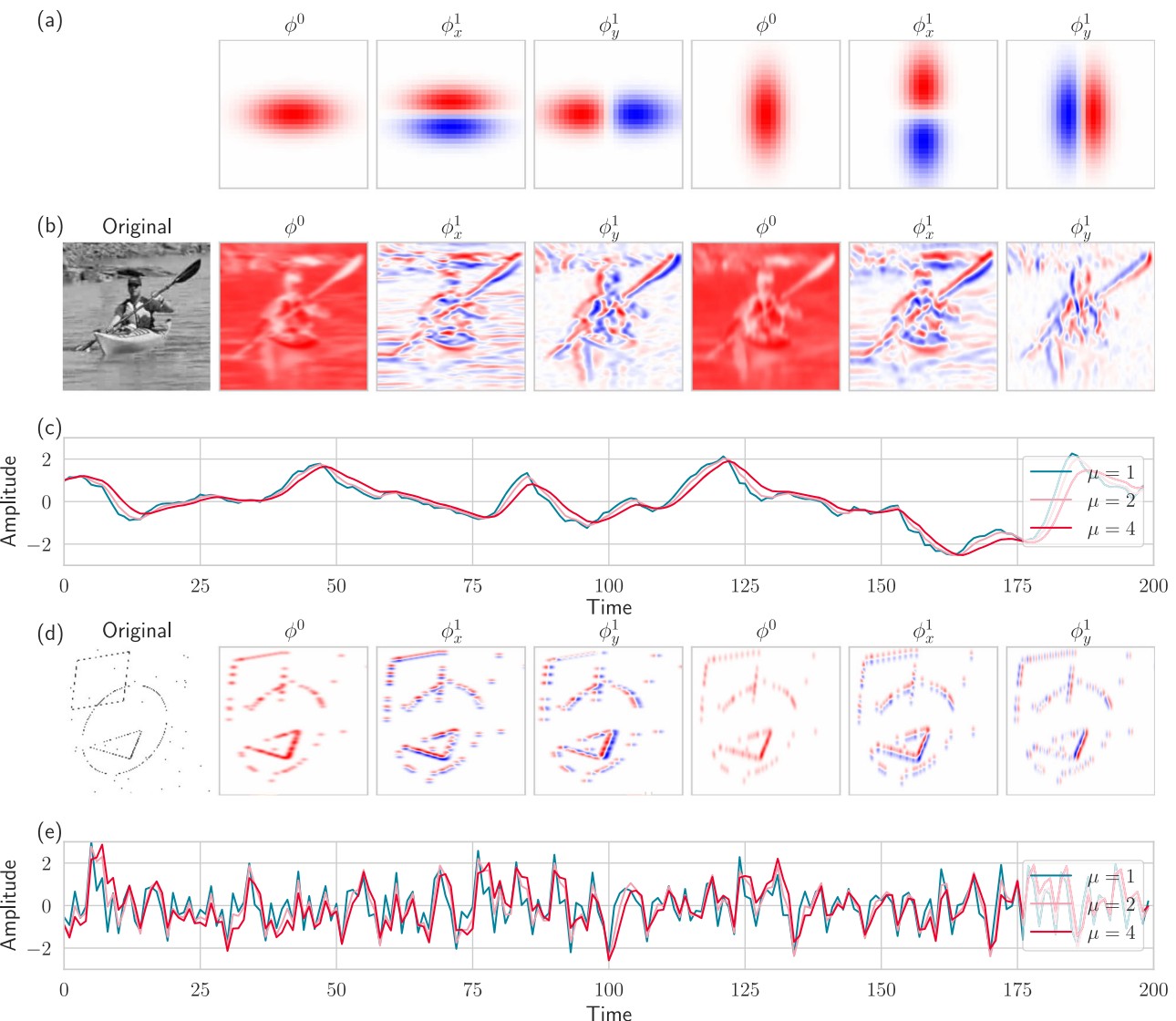

**Fig. 3 | Illustrations of spatial-temporal receptive field responses to natural video and event-based video. a** spatial receptive fields at angles 0° and 90° and their directional derivatives up to the first order. **b** a still frame from the UCF-101 dataset[S1] and the response for each of the kernels in (**a**) for $\mu = 1$. **c** averaged temporal traces from temporal receptive fields with three $\mu$ when applied to the UCF-101 video. **d** a still frame from our simulated event-based dataset and the response for each of the kernels in (**a**) for $\mu = 1$. **e** averaged temporal traces from temporal receptive fields with three $\mu$ values when applied to the video from (**d**). The sparsity and noise in the event-based video result in a much more irregular response over time.

they are not yet comparable in performance to methods used for training non-spiking deep networks[2,59,60]. To guide the training process for spiking networks, we propose to initiate the receptive fields with priors according to the idealized model for covariant spatio-temporal receptive fields. Following this idea, we used the purely spatial component of the spatio-temporal receptive fields by scale-normalized affine directional derivative kernels according to ref. 14, Equation (31) for $\gamma = 1$.

$$T_{\varphi^{m_1} \perp \varphi^{m_2}, norm}(x_1, x_2; \Sigma) = \sigma_\varphi^{m_1} \sigma_{\perp\varphi}^{m_2} \partial_\varphi^{m_1} \partial_{\perp\varphi}^{m_2} \left( g(x_1, x_2; s\Sigma) \right), \quad (5)$$

where

- $g(x_1, x_2; s\Sigma)$ is a Gaussian kernel with spatial scale parameter $s > 0$ and a positive definite covariance matrix $\Sigma$,
- $\partial_\varphi = \cos\varphi\, \partial_{x_1} + \sin\varphi\, \partial_{x_2}$ and $\partial_{\perp\varphi} = -\sin\varphi\, \partial_{x_1} + \cos\varphi\, \partial_{x_2}$ denote directional derivatives in directions $\varphi$ and $\perp\varphi$ parallel to the two eigendirections of the spatial covariance matrix $\Sigma$,
- $m_1$ and $m_2$ denote the orders of spatial differentiation, and

- $\sigma_\varphi = \sqrt{\lambda_\varphi}$ and $\sigma_{\perp\varphi} = \sqrt{\lambda_{\perp\varphi}}$ denote spatial scale parameters in these directions, with $\lambda_\varphi$ and $\lambda_{\perp\varphi}$ denoting the eigenvalues of the spatial covariance matrix $\Sigma$.

This receptive field model directly follows the spatial component in the covariant receptive field model detailed in the Section Joint covariance under geometric image transformations, while restricted to the special case when the image velocity parameter $v = 0$, and complemented with spatial differentiation, to make the receptive fields more selective. Receptive fields from this family have been demonstrated to be very similar to the receptive fields of simple cells in the primary visual cortex (see Figs. 16 and 17 in ref. 14). For the temporal scales, we initialize logarithmically distributed time constants in the interval [1, 4] for the leaky integrate-and-fire neurons, as described in the Methods Section called Model architecture and training.

Figure 3a visualizes a subset of the spatial receptive fields that we used in the experiments below, along with responses to both frame-based and event-based video. The panels (b) and (d) visualize the results

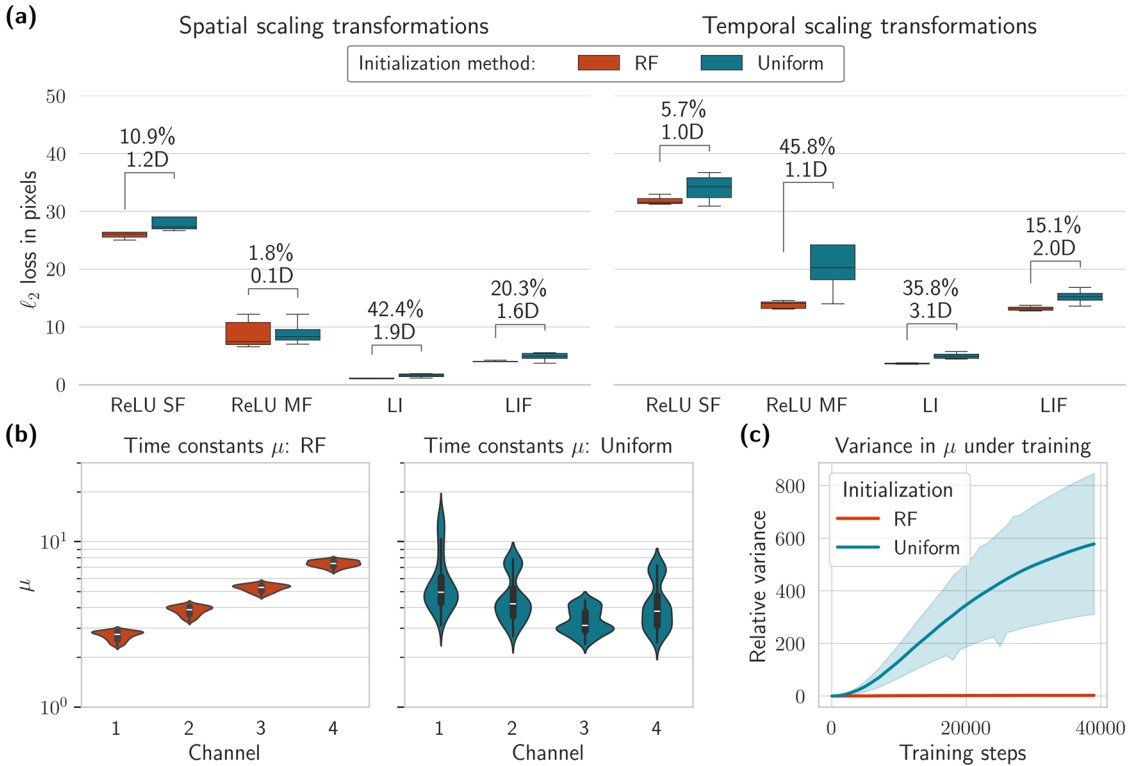

**Fig. 4 | Experimental results for event-based shapes subject to spatial or temporal scaling transformations across 5 training runs. a** Accuracy results for spatial (left) and temporal (right) scaling operations across four different models: ReLU single frame (SF), ReLU multi-frame (MF), leaky integrator (LI), and leaky integrate-and-fire (LIF). Each model is initialized either according to our receptive field theory (RF) or uniform white noise in the same parameter domain. The numbers above the distributions shows the difference in performance in percentage and the effect size (Cohen's d) of the initialization scheme. The whiskers capture the entire distribution, and the boxes indicate quartiles around the mean. Note the different y axes in the plots. **b** The distributions of the time constants $\mu$ in each temporal scale channel for the leaky integrator models trained on the temporal scaling task. **c** Relative variance of the time constants $\mu$ for the leaky integrator models as training for the temporal scaling task progresses, averaged across all channels. The shaded area indicates a 95% confidence interval.

of convolving the spatial receptive fields with conventional image data and event data, respectively, and the panels (c) and (e) show how the temporal traces change according to the temporal scale. When the temporal scale increases, the delay increases and effectively smooths out the signal. Striking a balance between the spatial and temporal scales is crucial for the model to be able to track the shapes in the event-based video data at appropriate temporal resolutions: too short temporal scales will produce imprecise and noisy predictions, while too long temporal scales will result in delayed and smeared-out responses.

### Experimental results

We proceed by experimentally measuring the effect of the initialization scheme in an event-based object tracking task. We built a 3-layer convolutional model whose parameters are configured either according to our receptive field (RF) theory or according to the default, uniform initialization scheme. Three kinds of activation functions are used: leaky integrators (LI), leaky integrate-and-fire (LIF) neurons, and stateless ReLU for a comparable baseline. Since LI and LIF models integrate signal over time and ReLU models do not, ReLU models are disadvantaged when processing temporal information such as sparse events. To compensate for the lack of state in ReLU models, we introduce a ReLU variant that operates on eight previous visual frames (multi-frame; MF), as opposed to the other single-frame (SF) models. This effectively provides the model with a temporal memory and cancel out much of the advantage of the LI and LIF models (see Section Model architecture and training for details).

Figure 4 shows the results when training our model to track sparse and event-based shapes subject to scaling transformations in space (left) and time (right) (see the Methods section Dataset for details on the task). Both in the spatial and temporal scaling scenarios, the effect of our receptive field (RF) initialization scheme is significant: with Cohen's d values of ≥1.9 for the leaky integrator (LI) models and ≥1.6 for the leaky integrate-and-fire models (LIF) (see Methods section Initialization effect size for details on the metric). In terms of raw performance, LI models initialized with RF priors achieve the lowest $L_2$ loss at 1.13 pixels for spatial scaling with a 42.4% improvement compared to uniform initialization. An even more dramatic improvement is seen in temporal scaling: 35.8% for RF vs. 41.6% for uniform initialization. The effect is still significant for the leaky integrate-and-fire model in the spatial case with a 20.3% improvement, but more certain in the temporal scaling task, despite being less pronounced (15.1%). The lower $L_2$ loss values for the spatial task across all models indicate that this task is easier to solve than temporal scaling. However, the effect of imbuing models with spatio-temporal receptive field priors is markedly stronger in the temporal scaling transformation task.

The initialization effect is less pronounced in the ReLU models (Cohen's d of 1.2 for SF and 0.1 for MF in spatial scaling). Since the gradient landscape is much simpler in ReLU models than spiking models[61], backpropagation should find more optimal parameter distributions as training progresses. Despite that, the ReLU models fall behind in overall performance. The single frame (SF) model is likely struggling with the highly sparse signals, as shown in Fig. 8, of which the model sees only a single frame per timestep. This is evidenced by the increased performance of the multi-frame model (MF), because the only difference to the SF model is the simultaneous exposure to eight frames extending backwards in time. Therefore, the increased performance must be due to the MF model's ability to correlate signals across frames. Even with multiple frames, however, the ReLU models

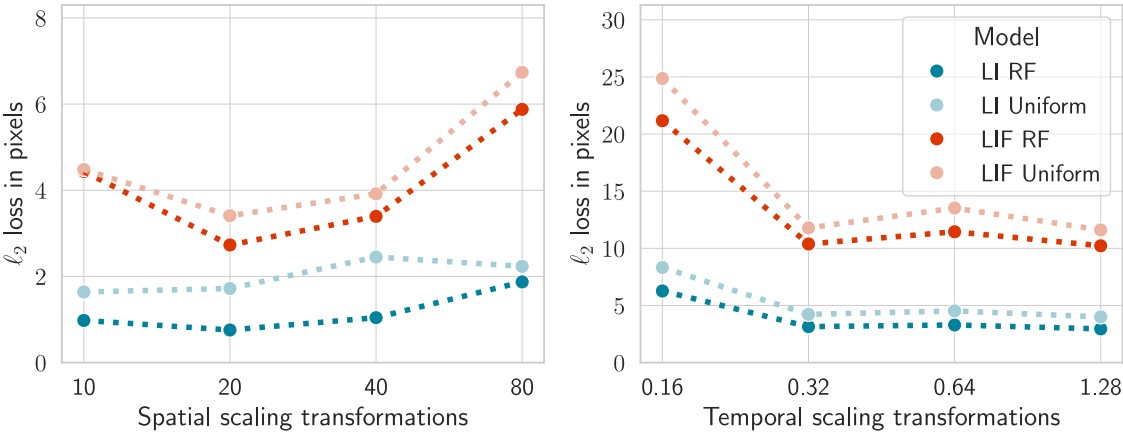

**Fig. 5 | The performance of the trained LI and LIF models when subject to unseen testing data at different scales.** Note the different *y*-axes.

are clearly outperformed by the leaky-integrator or leaky integrate-and-fire models, neither of which integrates more than one frame per timestep. This suggests that the temporal dynamics inherent in LI and LIF neurons provide a fundamental advantage in processing temporally scaled inputs compared to static ReLU units, even when the latter have access to multiple frames.

We additionally plotted the distributions of the time constants $\mu$ across different channels during training for the leaky integrator models in Fig. 4b. As expected, the RF initialization scheme provides a more stable distribution of time constants, even though the time constants are optimized with backpropagation through time, as with all the other model parameters. The uniformly initialized time constants are more spread out, as expected, but appear to converge around similar time constants $\mu$ as the RF models, indicating that the initialized values provide an optimal starting point in the parameter space. This stability is further confirmed in Fig. 4c, where we plot the variance of $\mu$ relative to their initial values. The time constants in the uniform models show dramatically increasing variance (up to 800 relative variance units) as training progresses, while RF-initialized models maintain near-zero variance throughout training. The 95% confidence intervals (shaded regions) demonstrate the consistency of this finding across multiple training runs, highlighting the reliability of our RF initialization approach.

Models that are provably covariant to scaling transformations should robustly represent and generalize to other scales[12,62]. To study the generalization across varying spatial and temporal scales, we ask the trained models to infer object positions at individual transformational scales.

Figure 5 shows the results for spatial and temporal scaling transformations for the leaky integrator and leaky integrate-and-fire models. The models initialized with spatio-temporal receptive fields are slightly more flat and, therefore, less sensitive to transformations in the data. The spatial models perform relatively worse at the largest scale, which can be explained by the fact that objects that are 80 pixels in diameter are difficult for small convolutional models to capture because they do not fit in any single kernel. Conversely, the temporal models struggle at small velocities. This can be attributed to the fact that the data is extremely sparse when the velocity is low, since the slow rate of movement does not trigger any pixel changes, as described in Section Dataset and shown in Fig. 8.

Both settings—the large spatial scale and the slow temporal velocities—are detrimental to the leaky integrate-and-fire model because they rely on non-zero inputs to cross the firing threshold. Without enough activity, it will never output any spikes, which makes the training even harder. The relatively worse performance compared to the leaky integrator model can additionally be explained by the complicated gradient landscape introduced by the surrogate gradients.

It is worth mentioning that receptive fields inherently provide a normalizing effect through their use of scale-normalized derivatives, as described in Section Temporal scale covariance for leaky integrators and leaky integrate-and-fire models[12]. This normalization property bears some conceptual similarity to batch normalization, a popular technique in deep learning that normalizes input signals and mitigates distributional shifts during training[63]. To investigate potential interactions between our RF initialization and explicit normalization techniques, we conducted additional experiments with batch normalization layers inserted before each temporal activation layer in the model (detailed in the Supplementary Material, Section F). As expected, batch normalization reduces the performance gap between RF and uniform initialization schemes, suggesting that part of the advantage conferred by RF initialization comes from its inherent normalizing properties. In addition, we observed that LIF models particularly benefit from batch normalization compared to their performance without it, likely because the normalization helps maintain more consistent non-zero inputs (and, therefore, gradients) throughout training.

## Discussion

We presented a theoretically well-founded model for covariant spatio-temporal receptive fields under geometric image transformations and temporally scaled signals, involving leaky integrator and leaky integrate-and-fire neuron models over time. We hypothesized that the initialization of spiking event-driven deep networks with priors from idealized receptive fields improves the training process. Our hypothesis was tested in a neural network on a coordinate regression event-based vision task. The experiments confirmed our theoretical prediction in that correctly initialized spatio-temporal fields significantly improve the performance of event-based networks.

We set out to study principled methods of computation in neuromorphic systems and argue that our findings carry three important implications. Firstly, we conceptually extended the scale-space framework to biologically inspired and spiking primitives, which provides an exciting direction for principled computation in neuromorphic systems. This establishes a direct link to the large literature around scale-space and conventional computer vision in the context of neuromorphic computing, such as covariant geometric deep learning, which have been successful in computer vision[16,29], polynomial basis expansions for memory representations, known as state space models[64,65], and reinforcement learning[30]. In addition, scale-space representations have been related to computation in biological vision systems and the primary visual cortex[14]. The covariance properties in event-based computation align well with experimental findings around time-invariant neural processing[39,66], episodic memory spectra[67], intrinsic coding of time scales in neural computation[56,68], and heterogeneous distributions of time constants in spiking networks[37]. As a

second implication, we uncover and exploit powerful intrinsic computational functions of neural circuits. Since the network architecture builds on biologically-inspired primitives and operates without temporal averaging and with asynchronous signals, it is an ideal candidate for implementation in neuromorphic and biophysical systems, possibly via the Neuromorphic Intermediate Representation[69]. Third, the immediate practical value of our approach relates to event-based vision tasks and signal processing in real-time settings, in particular when processing data at the edge, where resources are scarce and necessarily time-causal. Competing with the performance of ANNs in event-based vision tasks is a crucial step towards the serious use of neuromorphic systems for real-world applications.

This work provided an ideal setting with provable mathematical results that was subsequently demonstrated in a single network for carefully chosen covariance properties. As such, the size of the example network and the limited exploration of hyperparameters restrain the generalization of the method to other networks and tasks without further studies. The theoretical framework operates in the continuous domain, and some assumptions about the distribution and discretization of the spatio-temporal receptive fields may impact model performance. While we argue that the demonstrations here provide a promising step for event-based vision tasks, the findings call for further experiments to understand how the theoretical guarantees relate to other settings, including different hyperparameters, alternative geometries, and real-life datasets.

Since we base our theoretical findings on the spike-response model, which is powerful enough to describe numerous other neuron models, an interesting avenue for future work is to explore the generalization of our approach to other neuron models, such as the Hodgkin-Huxley model[70], the Izhikevich model[71], or the Fitzhugh-Nagumo model[72].

The main goal of this work was to establish a theoretical framework for spatio-temporal receptive fields that links computer vision and biologically-inspired computational vision processing. Theory tells us that systems endowed with spatio-temporal receptive fields similar to the ones presented in this paper provide more compact and efficient networks. Our findings show that the initialization of spiking event-driven deep networks with priors from idealized receptive fields improves the training process. While more work is needed to establish the generalization of the approach, we believe that our findings are a step towards more principled computation for vision processing in neuromorphic systems.

## Methods
### Covariance properties of spatio-temporal receptive fields
Consider a composition of the spatio-temporal image transformations defined in Section Joint covariance under geometric image transformations of the form

$$x' = A(x + u\,t), \tag{6}$$

$$t' = S_t\,t, \tag{7}$$

for two video sequences $f'$ and $f$ that are related according to $f'(x', t') = f(x, t)$. Further, define the spatio-temporal scale-space representations $L'$ and $L$ of $f'$ and $f$, respectively, by convolution of $f$ and $f'$ with the spatio-temporal smoothing kernels according to ref. 22

$$T(x, t;\ \Sigma, \tau, v) = g(x - v\,t;\ \Sigma)\,h(t;\ \tau), \tag{8}$$

where $\Sigma$ is a spatial covariance matrix that determines the spatial shape of the spatial Gaussian kernel $g(x;\ \Sigma)$, $v$ denotes an image velocity that specifies the velocity-sensitivity properties of the spatio-temporal

$$
\begin{array}{ccc}
L(x,t;\ \Sigma,\tau,v) & \xrightarrow[x'=A(x+ut)]{} & L'(x',t';\ \Sigma',\tau',v') \\
\uparrow & \Sigma'=A\,\Sigma\,A^T & \uparrow \\
& t'=S_t\,t & \\
\star\ T(x,t;\ \Sigma,\tau,v) & \tau'=S_t^2\,\tau & \star\ T(x',t';\ \Sigma',\tau',v') \\
& v'=(A\,v+u)/S_t & \\
\mid & & \mid \\
f(x,t) & \xrightarrow[\substack{x'=A(x+ut) \\ t'=S_t\,t}]{} & f'(x',t')
\end{array}
$$

**Fig. 6 | Commutative diagram for convolutions with the spatio-temporal kernel $T(x, t;\ \Sigma, \tau, v)$ according to Equation (8), which obeys joint covariance under compositions of spatial affine transformations, Galilean transformations, and temporal scaling transformations.** The commutative property implies that the order of transformations and spatio-temporal kernel applications is interchangeable.

receptive fields over joint space-time, and $\tau$ denotes the temporal scale of a scale-covariant temporal window function $h(t;\ \tau)$.

Then, these spatio-temporal image representations over the two spatio-temporal image domains are related according to

$$L'(x', t';\ \Sigma', \tau, 'v') = L(x, t;\ \Sigma, \tau, v), \tag{9}$$

provided that the parameters $(\Sigma, \tau, v)$ and $(\Sigma', \tau', v')$ of the receptive fields in the two domains, respectively, are related according to

$$\Sigma' = A\,\Sigma\,A^T, \tag{10}$$

$$\tau' = S_t^2\,\tau, \tag{11}$$

$$v' = (A\,v + u)/S_t. \tag{12}$$

This result can be proved by composing the individual image transformations in cascade, as treated in Lindeberg[15]. Figure 6 illustrates the commutative property of the transformations and the capturing mechanism: if an image is transformed (horizontal arrows) and then captured by a scale-space representation (vertical arrows), the result is the same as if we first capture the image and then transform it.

**Relationship under geometric image transformations.** Lindeberg and Gårding[73] established a relationship between two scale-space representations under linear affine transformations[13]. Concretely, if two spatial images $f(x)$ and $f'(x')$ are related according to $x' = A\,x$, and provided that the two associated covariance matrices $\Sigma$ and $\Sigma'$ are coupled according to ref. 73, Eq. (16),

$$\Sigma' = A\,\Sigma\,A^T \tag{13}$$

Then two spatially smoothed scale-space representations, $L$ and $L'$, are related according to,

$$L(x;\ s, \Sigma) = L'(x;\ s, \Sigma') \tag{14}$$

where $s$ parameterizes scale. When scaling from $s$ to $s'$ by a spatial scaling factor of $S_x$, we can relate the two scales by $s' = S_x^2\,s$ in the following scale-space representations[15], Eqs. (35) & (36)

$$L(x;\ s, \Sigma) = L'(x;\ s', \Sigma) \tag{15}$$

We exploit these relationships to initialize our spatial and temporal receptive fields by sampling across the parameter space for the spatial affine and temporal scaling operations.

For the spatial transformations, we sample the space of covariance matrices for Gaussian kernels parameterized by their orientation,

scale, and skew up to their second derivative. For computing directional derivatives from the images smoothed by affine Gaussian kernels, we apply compact directional derivative masks to the spatially smoothed image data, according to Data Availability in ref. 74, which constitutes a computationally efficient way of computing multiple affine-Gaussian-smoothed directional derivatives for the same input data. Examples of spatial Gaussian derivative kernels and their directional derivatives of different orders are shown in Fig. 3a. Combined with the convolutional operator, this guarantees covariance under natural image transformations according to Equation (14) in ref. 15.

To handle temporal scaling transformations, we choose time constants according to a geometric series, which effectively models logarithmically distributed memories of the past[56], Eqs. (18–20)

$$\tau_k = c^{2(k-K)} \tau_{max},$$ (16)

where $c$ is a distribution parameter that we set to $\sqrt{2}$, $K$ is the number of scales, and $\tau_{max}$ is the maximum temporal scale. Further details are available in Supplementary Material Section B.1.

This logarithmic space of scales guarantees time-causal self-similar treatment over temporal image scaling operations[15,57].

## Temporal scale covariance for leaky integrator models
The generalized Gaussian derivative model for receptive fields[14,22] defines a spatio-temporal scale-space representation

$$L(x, y; \Sigma, \tau, v) = \int_{\xi \in \mathbb{R}^2} \int_{u \in \mathbb{R}} T(\xi, u; \Sigma, \tau, v) f(\xi, u) \, d\xi \, du$$ (17)

over joint space-time ($x \in \mathbb{R}^2, t \in \mathbb{R}$) by convolving any video sequence $f(x, t)$ with a spatio-temporal convolution kernel of the form (9)

Previous Gaussian derivative models have exclusively been based on linear receptive fields, as detailed in Section 4 and in the Supplementary Materials Section B. Gaussian kernels are symmetric around the origin and, therefore, impractical to use as temporal kernels in real-time situations, because they violate the principle of temporal causality. Lindeberg and Fagerström[57] showed that one-sided, truncated exponential temporal scale-space kernels constitute a canonical class of temporal smoothing kernels, in that they are the only time-causal kernels that guarantee non-creation of new structure, in terms of either local extrema or zero-crossings, from finer to coarser levels of scale:

$$h_{exp}(t, \mu) = \begin{cases} \frac{1}{\mu} e^{-t/\mu} & t > 0, \\ 0 & t \leqslant 0, \end{cases}$$ (18)

where the time constant $\mu$ represents the temporal scale parameter corresponding to $\tau = \mu^2$ in (8).

Coupling $K$ such temporal filters in parallel, for $k \in [1, K]$, yields a temporal multi-channel representation in the limit when $K \to \infty$:

$$h_{composed}(\cdot; \tau_k) = h_{exp}(\cdot; \mu_k).$$ (19)

A specific temporal scale-space channel representation $L$ for some signal $f$[57], Eq. (14) can be written as

$$L(t; \mu) = \int_{u=0}^{\infty} f(t - u) h_{exp}(\cdot; \mu) \, du.$$ (20)

Turning to the domain of neuron models, a first-order integrator with a leak has the following integral representations for a given time $t$, time constant $\mu$ and some time-dependent input $I$[58], Eq. (1.11):

$$u(t; \mu) = \int_0^{\infty} \frac{1}{\mu} e^{-\xi/\mu} I(t - \xi) \, d\xi.$$ (21)

This precisely corresponds to the truncated exponential kernel in (18). Inserting into (20), we apply the temporal scaling operation $t' = S_t t$ for a temporal scaling factor $S_t > 0$ to the temporal input signals $f'(t') = f(t)$:

$$
\begin{aligned}
L(t'; \mu') &= \int_{u'=0}^{\infty} f'(t' - u') \frac{1}{\mu'} e^{-t'/\mu'} \, du' \\
&\overset{1}{=} \int_{u=0}^{\infty} f'(S_t(t - u)) \frac{1}{S_t \mu} e^{-S_t t/S_t \mu} S_t \, du \\
&= \int_{u=0}^{\infty} f(t - u) \frac{1}{\mu} e^{-t/\mu} \, du \\
&= L(t; \mu),
\end{aligned}
$$ (22)

where the step (1) sets $u' = S_t u$, $du' = S_t \, du$, and $\mu' = S_t \mu$. This establishes temporal scale covariance for a single temporal filter $T(t; \tau) = h_{exp}(t; \mu)$, while a set of temporal filters will be scale-covariant across multiple scales, provided the scales are logarithmically distributed[15], Section 3.2.

## Temporal scale covariance for leaky integrate-and-fire models
Continuing with the thresholded model of the first-order leaky integrator equation, we turn to the Spike Response Model defined in Section Spike response model below[58]. This model generalizes to numerous neuron models, but we will restrict ourselves to the leaky integrate-and-fire equations, which can be viewed as the composition of three filters: a membrane filter $\kappa$, a threshold filter $H$, and a membrane reset filter $\eta'$. In the case of the leaky integrate-and-fire model, we know that the resetting mechanism, as determined by $\eta'$, depends entirely on the spikes, and we can decouple it from the spike function $\Gamma$. If we further assume that the membrane reset follows a linear decay, governed by a time constant $\mu_r$, of the form described in Equation (30) in the Methods section, we arrive at the following formula for the leaky integrate-and-fire model:

$$z(t) = -\theta_{thr} e^{-(t-t_f)/\mu_r} + \int_0^{\infty} \kappa(s) I(t-s) \, ds.$$ (23)

Considering a temporal scaling transformation $t' = S_t t$ and $t'_f = S_t t_f$ for the temporal signals $f'(t') = f(t)$ in the concrete case of $\kappa = h_{exp}$, we follow the steps in Equation (22):

$$
\begin{aligned}
L(t'; \mu', \mu'_r) &= -\theta_{thr} e^{-(t'-t'_f)/\mu'_r} + \int_{z'=0}^{\infty} f'(t' - z') \frac{1}{\mu'} e^{-t'/\mu'} \, dz' \\
&\overset{1}{=} -\theta_{thr} e^{-S_t(t-t_f)/S_t\mu'_r} + \int_{z=0}^{\infty} f'(S_t(t-z)) \frac{1}{S_t \mu} e^{-S_t t/S_t \mu} S_t \, dz \\
&= -\theta_{thr} e^{-(t-t_f)/\mu_r} + \int_{z=0}^{\infty} f(t-z) \frac{1}{\mu} e^{-t/\mu} \, dz \\
&= L(t; \mu, \mu_r).
\end{aligned}
$$ (24)

This shows that a single leaky integrate-and-fire filter $z$ is scale covariant over time, and can be extended to a spectrum of temporal scales as above. Combined with the spatial affine covariance as well as Galilean covariance from Equation (9), this finding establishes spatio-temporal covariance to spatial affine transformations, Galilean transformations and temporal scaling transformations in the image plane for leaky integrators and leaky integrate-and-fire neurons.

## Spiking neuron models
We use the leaky integrate-and-fire formalization from Lapicque[32,75], stating that a neuron voltage potential $u$ evolves over time, according to its time constant, $\mu$, and some input current $I$

$$\mu \dot{u} = -u + I$$ (25)

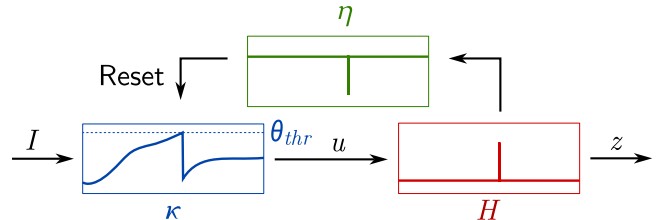

**Fig. 7 | The spike response model recasts the leaky integrate-and-fire model as a composition of three filters: a membrane filter in blue ($\kappa$), a threshold filter in red ($H$), and a reset filter in green ($\eta$).** The graphs inside the filters depict their respective outputs over time using the kernels in (28), given some input signal, $I$.

with the Heaviside threshold function parameterized by $\theta_{threshold}$

$$z(t) = \begin{cases} 1 & u >= \theta_{threshold} \\ 0 & u < \theta_{threshold} \end{cases} \tag{26}$$

and a jump condition for the membrane potential, which resets to $\theta_{reset}$ when the threshold is crossed

$$u(t) = \begin{cases} \theta_{reset} & z(t) = 1 \\ u(t) & z(t) < 1 \end{cases} \tag{27}$$

The firing part of the leaky integrate-and-fire model implies that neurons are coupled solely at discrete events, known as spikes.

**Spike response model.** The Spike Response Model describes neuronal dynamics as a compositions of filters[58], a membrane filter ($\kappa$), a threshold filter ($H$), and a membrane reset filter ($\eta$) shown in Fig. 7. Consider a reset filter ($\eta'$) applied to a spike generating function ($\Gamma$) and a subthreshold integration term ($\kappa$) given some input current ($I$):

$$u(t) = \int_0^\infty \eta'(s)\,\Gamma(t-s) + \kappa(s)\,I(t-s)\,ds \tag{28}$$

The reset filter, $\eta'$, is the resetting mechanism of the neuron, that is, the function dominating the subthreshold behavior immediately after a spike. We now recast the reset mechanism as a function of time ($t$), where $t_f$ denotes the time of the previous spike

$$u(t) = \eta(t - t_f) + \int_0^\infty \kappa(s)I(t-s)ds \tag{29}$$

Following[58], Eq. (6.33), we define $\eta$ more concretely as a linearized reset mechanism for the LIF model

$$\eta(t - t_f; \mu_r) = -\theta_{thr}\,e^{-(t-t_f)/\mu_r} \tag{30}$$

which describes the after-spike effect, decaying at a rate determined by $\mu_r$. Immediately following a spike, $e^0 = 1$ and the kernel corresponds to the negative value of $\theta_{thr}$ at the time of the spike, effectively resetting the membrane potential by $\theta_{thr}$. In the LIF model, the reset is instantaneous, which we observe when $\mu_r \to 0$. Figure 7 shows the set of filters corresponding to the subthreshold mechanism in Equation (21) (which we know to be scale covariant from Equations (22)), the Heaviside threshold, and the reset mechanism in Equation (30). We arrive at the following expression for the subthreshold voltage dynamics for the spike response model

$$z(t) = -\theta_{thr}\,e^{-(t-t_f)/\mu_r} + \int_0^\infty \kappa(s)\,I(t-s)\,ds \tag{31}$$

where $\kappa$ can be replaced by $h_{exp}$ for the leaky integrate-and-fire model.

## Dataset

We simulate an event-based dataset consisting of sparse shape contours, as shown in Fig. 8, based on ref. 76. Events are generated by applying some transformation $A$ to the shapes and subtracting two subsequent frames. We then integrate those differences over time until for each pixel, until reaching a certain threshold and emitting events of either positive or negative polarity. This approximates the dynamics of event cameras, which emit discrete events when the intensity changes by a certain threshold[77]. To obtain sub-pixel accuracy, we perform all transformations in a high-resolution space of $2400 \times 2400$ pixels, which is downsampled bilinearly to the dataset resolution of $300 \times 300$.

We use three simple geometries, a triangle, a square, and a circle, that are translated with random motion sampled from the same uniform distribution. The shapes are positioned randomly in the image and oriented by a random, fixed angle. Two datasets are studied: one with varying spatial scales and one with varying temporal velocities. In the spatial scale dataset, the starting scale of the shape is logarithmically distributed from 10 to 80 pixels. In the temporal velocity dataset, the shapes scale logarithmically from $\pm 0.16$ to $\pm 1.28$ pixels. The velocities are normalized to the subsampled pixel space such that, for translational velocity, for example, a velocity of one in the $x$-axis corresponds to the shape moving one pixel in the $x$-axis per timestep. The velocity parameterizes the sparsity of the temporal scaling dataset, as seen in the bottom row in Fig. 8 panel, where the shape contours in the left panels are barely visible compared to the faster-moving objects in the right panels. To simulate noise, we add 5 ‰ Bernoulli-distributed background noise.

The data is extremely sparse, even with high velocities. A temporal velocity of 0.16 provides around 1‰ activations, while a scale of 1.28 provides around 3‰ activations, including noise. Furthermore, random guessing yields an average $\ell^2$ error of 152 pixels when both points are drawn uniformly from a $300 \times 300$ cube[78]. If one of the points is fixed to the center of the cube, the error averages to 76 pixels.

## Model architecture and training

Figure 9 shows our network architecture, which combines a distribution of spatial receptive fields expanded over 4 parallel spatio-temporal scale channels into a single convolutional block. A scale channel is defined as a sequence of spatio-temporal receptive fields represented as convolutional blocks, where each block consists of a spatial convolution, and an activation function (a temporal and causal convolution).

The model architecture is fixed in all the experiments, but four different activation functions are studied in the scale channel blocks: a leaky integrator (LI), a leaky integrate-and-fire model (LIF), a ReLU, and a ReLU with additional historical information. The first two such functions are the leaky integrator, coupled with a rectified linear unit (ReLU) to induce a (spatial) nonlinearity, and the leaky integrate-and-fire models. To provide baseline comparisons to stateless models, we additionally construct a stateless Single Frame (SF) model that uses ReLU activations instead of the temporal kernels, and a Multiple Frame (MF) model that operates on eight consecutive frames instead of one. We chose eight frames for the frame-based model since it compares to an exponentially decaying leaky integrator with $\mu = 2$, which, after 8 timesteps, only retains about 2% of the original signal, specifically $\exp(-\frac{8}{2}) = 0.0183\%$.

To convert the latent activation space from the fourth block into 2-dimensional coordinates on which we can regress, we apply a differentiable coordinate transformation layer introduced in ref. 79, that finds the spatial average in two dimensions for each of the three shapes, as shown in Fig. 9. Note that the coordinate transform uses leaky integrator neurons for both the spiking and non-spiking model architectures.

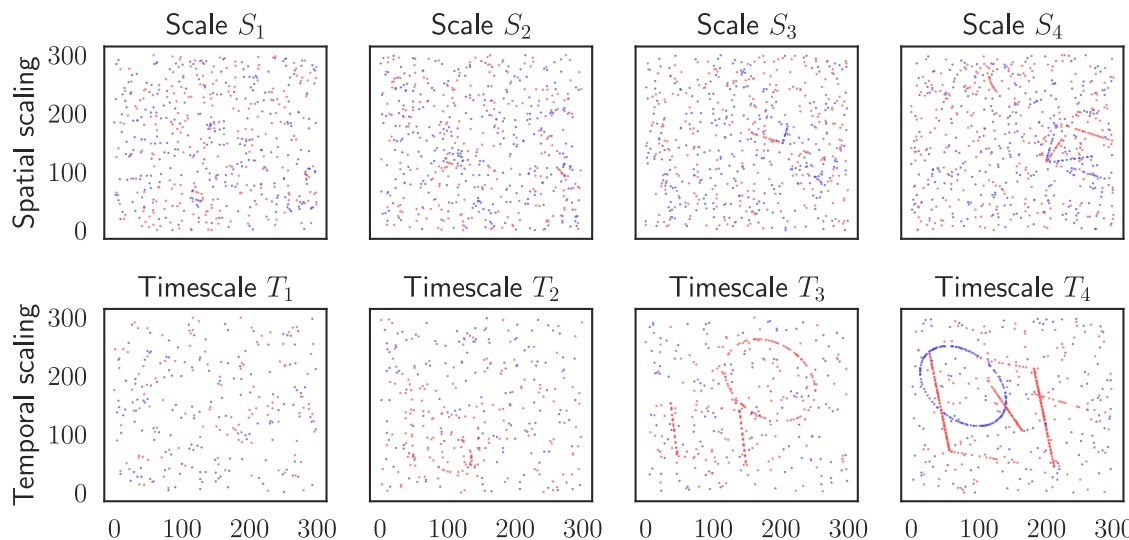

**Fig. 8 | Image snapshots from the two synthetic datasets used in this work.** The top row shows the four different spatial scaling, corresponding to shapes of 10, 20, 40, and 80 pixels. The bottom row shows the four different temporal velocities, corresponding to a change of 0.16, 0.32, 0.64, and 1.28 pixels per timestep. Both datasets are extremely sparse, with the shapes barely visible in the images. At larger temporal velocities, the shapes become more discernible.

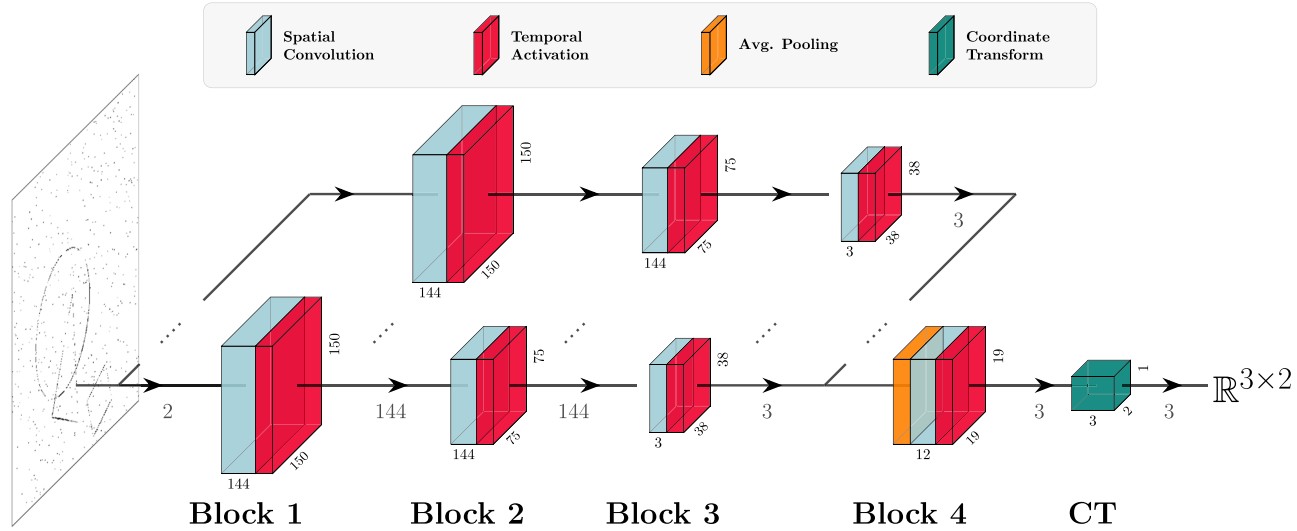

**Fig. 9 | The network architecture.** The first three convolutional blocks are parallelized into four different temporal scale channels that are sensitive to different temporal scales. Block 4 merges all the channels and produces three object-specific output channels that are transformation into coordinates in $\mathbb{R}^2$ in the Coordinate Transformation (CT) block. Those coordinates are compared to the labeled position of the sparse event-based shapes in the dataset. When initializing models according to theory, the first two blocks are configured as described in Section Parameter initialization below. Parameters in the remaining two blocks are uniformly initialized except the temporal activation time constant in Block 3, which is fixed to the fastest time constant.

**Training.** We train the models using backpropagation through time on datasets consisting of movies with 50 frames of 1 ms duration, with with 20 % of the dataset held for model validation. The full sets of parameters, along with steps to reproduce the experiments, are available in the Supplementary Material and online at https://github.com/jegp/nrf.

**Parameter initialization.** The spatial and temporal receptive fields that constitute the spatio-temporal blocks above are initialized by sampling over the space of all possible geometric image transformations according to Equations (10), (11), and (12). Figure 1 illustrates a sample of 4 × 4 different spatial scales $\sigma \in \{1, 2, 4, 8\}$ and temporal scales $\mu \in \{1, 2, 4, 8\}$, whereas Fig. 2 is restricted to three temporal scales ($\mu \in \{8, 4, 2\}$).

The geometric image transformations are modeled via spatial affine operations $A$, image velocities $v$, and constant shifts $u$, where the covariance matrices $\Sigma$ of the affine Gaussian kernels are influenced by the affine transformations $A$ according to Equation (10). We parameterize each receptive field according to the orientation, scale, and skew of the covariance matrix, as well as the $n$:th-order derivative of the receptive field, as described in Equation (5). The temporal transformation is of the form (11) and is controlled by the time constant $\tau$, determined from the time constants $\mu$ in the first-order integrators (21) and (23) according to $\tau = \mu^2$.

Ideally, we wish to measure the continuous change for all the parameters above, but we restrict the models to 144 spatial convolutional kernels per block and 4 temporal scales. The spatial kernels are linearly sampled over 4 orientations, 4 spatial scales, 3 skewness

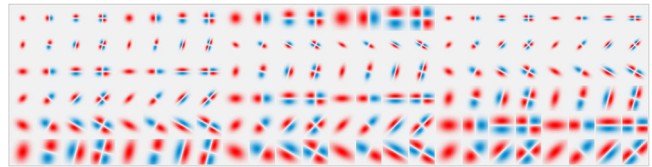

**Fig. 10 | Distribution of 144 spatial receptive fields for the convolutional blocks in the model at a high resolution.** The kernels in the deep network are initialized from this distribution, obtained by sampling over the space of spatial transformations and their derivatives. For the discrete implementation model, these continuous kernels are downsampled to $9 \times 9$ pixels.

values, and derivative orders of zeroth, first orders ($\delta_x$, $\delta_y$), and one second order derivative ($\delta_x y$), shown in Fig. 10. We sample the time constants according to the geometric series in Equation (16) with $K = 4$ scales and the distribution parameter $c = \sqrt{2}$. We uniformly initialize the spatial convolutional kernels in the third and fourth blocks to allow higher-order features to form that may be data-dependent and not captured by the theory.

**Initialization effect size.** To quantify the effect of the initialization, we measure the size of the effect on the model's performance, described as the difference in the mean loss values $\ell$ between the mean of the initialized $\ell_{\text{init}}$ and the mean of the uniformly initialized $\ell_{\text{uniform}}$ models. Since we are working with multiple evaluations of the same model, we correct for the spread of the distributions using the pooled standard deviation $\sigma_{\text{pool}}$:

$$\text{Effect size} = \frac{\ell_{\text{init}} - \ell_{\text{uniform}}}{\sigma_{\text{pool}}}. \tag{32}$$

The pooled standard deviation is calculated following[80], where $n_i$ is the number of samples in the $i$:th group,

$$\sigma_{\text{pool}} = \sqrt{\frac{(n_1 - 1)\sigma_1^2 + (n_2 - 2)\sigma_2^2}{n_1 + n_2 - 2}}, \tag{33}$$

and $\sigma^2$ is the variance of the samples defined by

$$\sigma^2 = \frac{\sum_{i=1}^{N}(n_i - 1)\sigma_i^2}{\sum_{i=1}^{N}(n_i - 1)}, \tag{34}$$

The standardized effect size indicates the number of standard deviations the mean of the initialized models is from the mean of the uniformly initialized models. More than one standard deviation implies a large and significant effect.

## Data availability
Training data used in the paper is generated using the event-based generator from[76] and can be recreated by following the instructions available at https://github.com/jegp/nrf and https://doi.org/10.5281/zenodo.16651826.

## Code availability
The network model and training code is available at https://github.com/jegp/nrf and https://doi.org/10.5281/zenodo.16651826.

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

## Acknowledgements

The authors gratefully acknowledge support from the EC Horizon 2020 Framework Program under Grant Agreements 785907 and 945539 (HBP) (J.E.P. and J.C.), the Swedish Research Council under contract 2022-02969 (T.L.), and the Danish National Research Foundation grant number P1 (J.E.P.). The computations were enabled by resources provided by the National Academic Infrastructure for Supercomputing in Sweden (NAISS), partially funded by the Swedish Research Council through grant agreement no. 2022-06725.

## Author contributions

Conceptualization: J.E.P., J.C., and T.L. Theory developments: J.E.P. and T.L. Dataset generation: J.E.P and J.C. Experiments: J.E.P. Analysis: J.E.P., J.C., and T.L. Writing: J.E.P., J.C., and T.L.

## Funding

## Competing interests

The authors declare no competing interests.
