## [Transparent Peer Review file · Nature Communications]

Covariant spatio-temporal receptive fields for spiking neural networks

Corresponding Author: Mr Jens Pedersen

Version 0:

Reviewer comments:

Reviewer #1

(Remarks to the Author)
Summary

This paper presents a novel approach to initializing spatio-temporal receptive fields in spiking neural networks (SNNs) for event-based vision tasks. The authors extend the generalized Gaussian derivative model to spatio-temporal receptive fields in the context of neuromorphic computing for the (leaky) integrate-and-fire model. By leveraging principles from scale-space theory and affine transformations, the authors aim to enhance the robustness and efficiency of neuromorphic systems. The proposed method integrates spatial Gaussian kernels and temporal (leaky) integrator models, ensuring covariance under affine and temporal scaling transformations. They provide a principled way of initializing the receptive fields, which are then learned using backpropagation. The results demonstrate significant improvements in L2 loss, highlighting the potential of this approach.

Review

1. Introduction

- The introduction could benefit from a more detailed discussion of how the proposed method specifically addresses the identified gaps in current models. Additional motivation explaining why covariance to spatial affine and temporal scaling transformations is useful would be helpful. Including practical examples or use cases would strengthen the motivation.

2. Related Work

- Some references are mentioned without sufficient detail on their relevance to the proposed method. A more critical analysis of these works would strengthen this section. For instance, in the sentence "Interestingly, the notion of scale space relates directly to sensory processing in biology, where scale-invariant representations are heavily used," more context and examples would improve clarity.

3. Methods

- The complexity of the mathematical notation might be inaccessible to readers without a strong background in mathematics. Including more intuitive explanations alongside the formal derivations would be beneficial.
- The section could benefit from visual aids or diagrams to illustrate the concepts and equations more clearly.

4. Initialization Scheme

- The description of the parameter space and sampling process is somewhat abstract. Including a step-by-step example with specific parameter values would clarify the process.
- The section could be more explicit about how the chosen parameters (orientations, scales, covariance matrices, time constants) impact the performance of the receptive fields.

5. Results

- The results section lacks detailed numerical data and specific examples. More analysis/comparisons and interpretation of the results would be desirable.
- Are the results robust to the choice of hyperparameters? This should be discussed and analyzed.
- Five runs to compute the standard deviation is fairly low; increasing the number of runs would provide more robust statistical insights.
- Testing the impact of increasing noise on performance would strengthen the robustness analysis.
- It would be beneficial to test the performances with more complex object shapes than triangles, squares, and circles.
- Adding acronyms in the figures to improve readability (e.g., SF: single frame, MF: multiple frames) would be helpful.

6. Discussion

- The discussion could be more critical, addressing the limitations of the study and potential challenges in real-world applications beyond synthetic datasets.
- The section does not sufficiently explore the computational cost of the proposed method, which is crucial for practical implementations, especially in the context of neuromorphic computing. For instance, is this model implementable in a neuromorphic processor?

7. Conclusion

- The conclusion briefly touches on future work but lacks specificity. It is too open and general. More concrete suggestions for next steps and how the community can build on these findings would be helpful.

Decision

Recommendation: Minor Accept with Major Revisions

- The paper presents a valuable contribution to the field, but it requires revisions to improve clarity, provide more detailed results, and offer a more critical discussion of the findings and their implications.

(Remarks on code availability)

Reviewer #2

(Remarks to the Author)

(Remarks on code availability)

Reviewer #3

(Remarks to the Author)

This work aims to understand the spatio-temporal computations of spiking neurons in visual processing. The model is first derived from first principles then applied to a leaky integrate-and-fire model neuron, demonstrating the advantage that temporal receptive fields have over standard models along with an advantage in the training process.

Considering the finding that initialisations with logarithmic scaling do better than uniform distributions, the authors may find the following publication relevant: Perez-Nieves et al. (2021) Neural heterogeneity promotes robust learning. Nat Commun 12, 5791. <https://doi.org/10.1038/s41467-021-26022-3>.

In general, the results presented are promising but brief and somewhat unconvincing. For example, the authors state:

"They also showed that the spatio-temporal receptive fields provided a clear advantage compared to baseline ANN models, despite the ANN having access to a complete history of recent inputs."

The LIF consistently performs worse than the LI and typically also worse than the ANNs although the relationship is less clear on the affine transformations - could the authors comment on this? In general, further demonstrations would be welcome to bolster the authors' conclusions about the advantages of spatio-temporal receptive fields.

Using the term "neuromorphic computing" in the title also suggests that neuromorphic hardware was used but I could not see the use of any in the methods. Although the term can apply to software, as it stands the title is a little misleading, so I would suggest changing it to "Covariant spatio-temporal receptive fields for spiking neural networks" to better match the scope of the work.

The paper addresses a very interesting topic and is mathematically detailed and rigorous. However, given the limited

results, it is perhaps better suited to a more specialist journal unless the advantages may be more thoroughly demonstrated.

There are also a number of minor grammatical and typographical errors which I list below.

L41: "...from the lens of..."  "...through the lens of..."

L124: "smothing"  "smoothing"

L179: $t'_f = S_t t'_f$  $t'_f = S_t t_f$

L185: integrator-and-fire neurons  integrate-and-fire neurons

L205-206: as described *in* Section 4.1 (add "in")

L208: and panels ~and~ (c) and (e) show *how* the temporal traces change (delete "and"; add "how")

L209: time scale. providing an implicit (replace the full stop with a comma)

Fig 2: "clipe"  "clip"

L223: "leaky integrate-and-fire integrator"  "leaky integrate-and-fire neuron"

L223-224: "particularly in the likely" (I'm not sure what "likely" means here)

L236: "and simple feed-forward architecture"  "and a simple feed-forward architecture"

L248: "naive"  "naïve"

L356: A091505 - I'm not sure what this is.

(Remarks on code availability)

The code is clear and well organised and freely available under version control at the provided GitHub URL. Considerable effort has gone into making a simple yet well-presented accompanying website to showcase the results. I did not attempt to run the code however as the README file did not contain sufficient instructions to run it and reproduce the paper's results and figures, nor were there many inline comments. The repository is also missing a requirements file specifying the exact library versions used (such that the exact computational environment and results may be precisely reproduced). Similarly, random seeds (if used) should be explicitly recorded in the README. There are references to using Jupyter notebooks so I recommend providing one which walks the user through the process of running the models and visualising the results presented in the paper, perhaps with a link to open the environment in mybinder.org.

Version 1:

Reviewer comments:

Reviewer #1

(Remarks to the Author)

The authors have correctly answered to the comments.

(Remarks on code availability)

Reviewer #2

(Remarks to the Author)

(Remarks on code availability)

Reviewer #3

(Remarks to the Author)

I believe the manuscript has been adequately revised regarding my points. There are however several typos to correct that I list below:

L42: "Covariance have gained popularity"  Covariance has?

L112: "More recently, scale-space theory has been enhanced the performance"  has enhanced

Fig. 1: The "100" and "0" on the x-axes are elided - please increase the spacing or adjust the font to prevent this.

Fig. 2: "When we scale three identical squares ~in~ at different scale velocities..."

Fig. 2b: Has no introduction in the caption.

Fig. 2b & c: Axes have no labels (or explanation in the caption)

L183: "over a set different"  over a set of different

L184: Missing full stop at the end of the sentence.

L255: ?

Fig. 3: Axes have no labels.

Fig. 3: "three μ values*" in two places in the caption.

L300: "smoothens"  smooths

(Remarks on code availability)

AUTHORS' RESPONSE TO EDITOR'S AND REVIEWERS' COMMENTS

REVIEWERS' COMMENTS

We thank the reviewers for their thoughtful and constructive feedback, which has significantly improved the quality and scientific rigor of our manuscript. We believe the revised paper now presents a more compelling demonstration of our theoretical framework and its practical applications.

Reviewer #1 (Remarks to the Author): Summary

This paper presents a novel approach to initializing spatio-temporal receptive fields in spiking neural networks (SNNs) for event-based vision tasks. The authors extend the generalized Gaussian derivative model to spatio-temporal receptive fields in the context of neuromorphic computing for the (leaky) integrate-and-fire model. By leveraging principles from scale-space theory and affine transformations, the authors aim to enhance the robustness and efficiency of neuromorphic systems. The proposed method integrates spatial Gaussian kernels and temporal (leaky) integrator models, ensuring covariance under affine and temporal scaling transformations. They provide a principled way of initializing the receptive fields, which are then learned using backpropagation. The results demonstrate significant improvements in L2 loss, highlighting the potential of this approach.

Thank you for your summary and appreciation of these main contributions in the paper.

Review

1. Introduction

- The introduction could benefit from a more detailed discussion of how the proposed method specifically addresses the identified gaps in current models. Additional motivation explaining why covariance to spatial affine and temporal scaling transformations is useful would be helpful. Including practical examples or use cases would strengthen the motivation.

We have added a paragraph in the introduction that motivates covariance for (event-based) computer vision and clarifies that the goal is theoretically optimal encodings. We further detail previous work that serves as concrete examples. Finally, we added references from the field of deep learning in the "Related work" section in the introduction to serve as additional motivating examples.

2. Related Work

- Some references are mentioned without sufficient detail on their relevance to the proposed method. A more critical analysis of these works would

strengthen this section. For instance, in the sentence “Interestingly, the notion of scale space relates directly to sensory processing in biology, where scale-invariant representations are heavily used,” more context and examples would improve clarity.

The paragraph in question has been clarified to state exactly how scale space relates to sensory processing in biology. Furthermore, we revisited references and claims of relationships to other papers in the introduction and discussion sections.

3. Methods

- The complexity of the mathematical notation might be inaccessible to readers without a strong background in mathematics. Including more intuitive explanations alongside the formal derivations would be beneficial. *The reviewer brings up a good point, and we decided to restructure the Result and Methods Section to (1) provide a much lighter and more accessible explanation, (2) briefly provide the theoretical results, and (3) refer to the Methods section and Supplementary Material for additional details, including proofs which has been moved from the main result text. The first part of the results section now explains the transformations our data are subject to and the covariance properties we are interested in, aided by two additional figures. Figure 1 visualizes the receptive fields with varying spatial and temporal scaling parameters. Figure 2 shows how the response of a receptive field co-varies as the signal changes in scale. The second part on the theoretical results now simply states the results with a few main equations, then referring to the Methods section and the Supplementary Material for additional details for the interested reader.*
- The section could benefit from visual aids or diagrams to illustrate the concepts and equations more clearly.

We agree that the amount of notation and lack of illustrations could require a certain effort for readers less familiar with in-depth mathematical analysis. To address this, we have:

- a. *Added a visualization (Figure 1 in the revised version) of receptive fields parameterized by spatial and temporal scaling parameters.*
- b. *Added a visualization (Figure 2 in the revised version) that demonstrates temporal scale covariance along with a detailed explanation. The result section now starts with a clarification of the geometric image transformations before we proceed with our contributions.*

4. Initialization Scheme

- The description of the parameter space and sampling process is somewhat abstract. Including a step-by-step example with specific parameter values would clarify the process.

We reworked the section on “Initializing deep networks with idealized receptive fields” to clarify, and provide a better intuition for, the necessary balance between spatial and temporal scales. Figures 1 and 2 exemplify two choices of receptive field parameterization: Figure 1 shows 16 different 4 spatial and temporal scales, where Figure 2 shows 3 temporal scales. We additionally added a section in the Methods section on “Parameter initialization” that provides a step-by-step example for both the spatial and temporal parameters.

- The section could be more explicit about how the chosen parameters (orientations, scales, covariance matrices, time constants) impact the performance of the receptive fields.

We addressed this comment in four ways: (1) first, we added a paragraph in the section on “Joint covariance under geometric image transformations” that explains the importance of aligning the spatio-temporal receptive field parameters with the transformations in the data. (2) Second, we updated Figure 3 to more clearly show how different kernels give different outputs (and, therefore, “select” for specific patterns). (3) Third, our experimental analysis on the temporal kernels (shown in Figure 4b and c) shows how the initialization changes the distribution. (4) Finally, Figure 5 visualizes the effect of the initialization scheme on the different scales in the spatial and temporal datasets.

5. Results

- The results section lacks detailed numerical data and specific examples. More analysis, comparisons, and interpretation of the results would be desirable.

We improved the presentation in the result section by (1) providing concrete examples for both spatial and temporal covariance in Figures 1 and 2, (2) restructuring the experiments to focus on individual questions around spatial and temporal covariance, (3) provide standardized effect sizes (Cohen’s d), (4) include analyses on the network performance and time constants, and (5) study the performance at individual transformational scales in Figure 5.

- Are the results robust to the choice of hyperparameters? This should be discussed and analyzed.

The question whether the results are robust to the choice of hyperparameters is an important one that deserves further investigation, but that we have not addressed in our study. While we conducted limited explorations and observed that network size affects the impact of our scale-space initialization approach, we did not find conclusive patterns sufficient for inclusion in this manuscript. Our present theory provides an ideal setting with provable mathematical results, but will necessarily have different impacts due to effects like quantization of the receptive fields (spatially and temporally), integration discretization, data distribution, and other parameters. We

argue that a proper treatment of this is a worthy endeavor, but should be regarded as a separate task, as a complement to the theory put forth in the submitted manuscript.

We have added an explicit discussion of these limitations in the revised manuscript's Discussion section, acknowledging this as an important area for future work.

- Five runs to compute the standard deviation is fairly low; increasing the number of runs would provide more robust statistical insights.

The reviewer brings a good point. Regrettably, the models are quite large and cumbersome to train due to the complexity of backpropagation-through time. Additionally, the combinatorial growth of the configurations, based on model type, initialization scheme, single frame, multi-frame, dataset, and batch normalization made it challenging to run additional models in time. However, increasing the statistical validity of the experimental result is an appropriate and important next step. This point has been added to the discussion.

However, we have added standardized effect sizes (Cohen's d) for each experiment, analyzed them in the result section, and discussed the ability of our method to generalize in the discussion. The approach is explained in the Method section "Initialization effect size".

- Testing the impact of increasing noise on performance would strengthen the robustness analysis.

We agree that it would be interesting to understand the effect of noise, particularly because noise is prevalent in event-based vision. In the revised experiments we have increased the noise level and added a significant amount of noise to our data (5% of the pixels) partly to understand whether our method is robust to noisy signals, partly to avoid overfitting. The ability of the proposed method to handle such high noise levels shows that the presented scheme has a clear robustness to noise. A full treatment of the effect of noise is an experimental endeavor. Doing it extensively, would require significantly more experiments ranging over significantly more datasets, and thus a significantly larger amount of computational work over variations of different noise levels. Following the argument above regarding the hyperparameters, we suggest postponing such studies for future work. As can be seen from the illustrations of the datasets in Figure 8 in the revised manuscript of the datasets used for the new experiments in the revised paper, the noise level in the baseline experiments is substantial.

- It would be beneficial to test the performances with more complex object shapes than triangles, squares, and circles.

While we fully agree with the reviewer that it would be beneficial to test more complex objects, the purpose of our work is to understand covariance properties for arbitrary shapes in a highly controlled

setting. We added the reviewer's point in the discussion to point out that future work should focus on this, along with more realistic event-based camera data.

- Adding acronyms in the figures to improve readability (e.g., SF: single frame, MF: multiple frames) would be helpful.

We have now added the acronyms in the caption and main text.

6. Discussion

- The discussion could be more critical, addressing the limitations of the study and potential challenges in real-world applications beyond synthetic datasets.

We expanded and clarified the list of limitations, which now details the lack of generalization, simplicity of the tasks, synthetic dataset, gradient training of spiking neurons, and small set of neuron models.

- The section does not sufficiently explore the computational cost of the proposed method, which is crucial for practical implementations, especially in the context of neuromorphic computing. For instance, is this model implementable in a neuromorphic processor?

The discussion section now includes a treatment on how this approach is a good fit for neural processing in general because all the processing is done in fully asynchronous neurons. We additionally describe how the method can translate to neuromorphic hardware using the Neuromorphic Intermediate Representation and that we consider it to be an important step towards competing with ANNs for (real-time) event-based vision pipelines.

7. Conclusion

- The conclusion briefly touches on future work but lacks specificity. It is too open and general. More concrete suggestions for next steps and how the community can build on these findings would be helpful.

We restructured the section to first discuss the study, the results, and their impact. The second part of the section details the limitations with concrete hypotheses, suggestions, and references for future work.

Decision: Recommendation: Minor Accept with Major Revisions

- The paper presents a valuable contribution to the field, but it requires revisions to improve clarity, provide more detailed results, and offer a more critical discussion of the findings and their implications.

Thank you for the appreciation. In the revised manuscript, we have now added a wider discussion about the findings, their implications, as well as suggestions for future work.

Reviewer #2 (Remarks to the Author):

I co-reviewed this manuscript with one of the reviewers who provided the listed reports. This is part of the *Nature Communications* initiative to facilitate training in peer review and to provide appropriate recognition for Early Career Researchers who co-review manuscripts.

Reviewer #3 (Remarks to the Author):

This work aims to understand the spatio-temporal computations of spiking neurons in visual processing. The model is first derived from first principles, then applied to a leaky integrate-and-fire model neuron, demonstrating the advantage that temporal receptive fields have over standard models along with an advantage in the training process.

Thank you for the appreciation.

Considering the finding that initializations with logarithmic scaling do better than uniform distributions, the authors may find the following publication relevant: Perez-Nieves et al. (2021) Neural heterogeneity promotes robust learning. *Nat Commun* 12, 5791. <https://doi.org/10.1038/s41467-021-26022-3>.

The publication is indeed relevant and has been added to the related work section and to the discussion.

In general, the results presented are promising but brief and somewhat unconvincing. For example, the authors state:

“They also showed that the spatio-temporal receptive fields provided a clear advantage compared to baseline ANN models, despite the ANN having access to a complete history of recent inputs.”

The result section has been reworked with (1) a clear exposition of covariance properties for spatio-temporal transformations with added effect sizes (Cohen’s d) and (2) experiments that clearly address isolated transformational covariance in space and time.

The experimental results now study separate properties carefully in a step-by-step manner. First, we address (1) the effect of the initialization scheme and the distribution of time constants. Second, we address the performance of the LI and LIF models across different scales to study their ability to generalize. Additionally, the introduction to the result section and the explanations of the models in the Methods section have been revised to better guide the reader through the experimental setup and findings.

The LIF consistently performs worse than the LI and typically also worse than the ANNs although the relationship is less clear on the affine transformations - could the authors comment on this? In general, further demonstrations would be welcome to bolster the authors’ conclusions about the advantages of spatio-temporal receptive fields.

In the new version of the manuscript with the revised model, the LIF model outperforms the ANN models. We present and analyze the performance of the model both in terms of their effect size (Cohen's d , Figure 4) and the performance for individual transformational scales (Figure 5).

Addressing the point on further demonstrations, we present the (theoretical) benefits of covariance properties in receptive fields in the beginning of the result section (Section 2.1 Joint covariance under geometric image transformations). Additionally, we restructured the experiments to disambiguate the relationship between the models and the impact of the suggested initialization scheme.

Using the term “neuromorphic computing” in the title also suggests that neuromorphic hardware was used but I could not see the use of any in the methods. Although the term can apply to software, as it stands, the title is a little misleading, so I would suggest changing it to “Covariant spatio-temporal receptive fields for spiking neural networks” to better match the scope of the work.

Thank you for the remark. While our goal is to implement the proposed methods on neuromorphic processors, which we have in our lab, we have not built those implementations yet. Therefore, we have changed the title of the paper according to your suggestion.

The paper addresses a very interesting topic and is mathematically detailed and rigorous. However, given the limited results, it is perhaps better suited to a more specialist journal unless the advantages may be more thoroughly demonstrated.

We appreciate the praise for the topic and rigor. We respectfully believe that Nature Communications is indeed the appropriate venue for this work for two key reasons: (1) First, our contribution extends beyond theoretical interest to demonstrate practical value. The initialization benefits we show for event-based vision models represent a promising step toward neuromorphic systems that can compete with conventional deep learning approaches. Our results provide concrete evidence that properly initialized spatio-temporal receptive fields significantly improve performance in event-based neural networks, which addresses a critical gap in the field. (2) Second, we address a fundamental challenge in neuromorphic computing: the lack of falsifiable computational models for event-driven systems. Our framework of scale-specific representations that preserve signal transformation structure provides both explanatory and predictive power for sparse event-based computations. This approach establishes principled mathematical foundations that can guide future neuromorphic hardware and algorithm design. We have strengthened the manuscript with additional figures, improved exposition of both theoretical and practical results, and a reworked introduction that highlights these broader impacts. We believe these changes make our contribution valuable and accessible to the readers of the special neuromorphic issue in Nature Communications.

There are also a number of minor grammatical and typographical errors which I list below.

- **L41**: "...from the lens of..." → "...through the lens of..."
- **L124**: "smothing" → "smoothing"
- **L179**: $t'_f = S_t t'_f$ → $t'_f = S_t t_f$
- **L185**: integrator-and-fire neurons → integrate-and-fire neurons
- **L205-206**: as described *in* Section 4.1 (add "in") -

L208: and panels _{and} (c) and (e) show *how* the temporal traces change (delete "and"; add "how")

L209: time scale. providing an implicit (replace the full stop with a comma) -

Fig 2: "clipe" → "clip" -

L223: "leaky integrate-and-fire integrator" → "leaky integrate-and-fire neuron"

- **L223-224**: "particularly in the likely" (I'm not sure what "likely" means here)

- **L236**: "and simple feed-forward architecture" → "and a simple feed-forward architecture"

- **L248**: "naive" → "naïve"

- **L356**: A091505 - I'm not sure what this is.

Thank you for observing these typing errors, which have now been corrected.

Reviewer #3 (Remarks on code availability):

The code is clear and well-organized and freely available under version control at the provided GitHub URL. Considerable effort has gone into making a simple yet well-presented accompanying website to showcase the results. I did not attempt to run the code, however, as the README file did not contain sufficient instructions to run it and reproduce the paper's results and figures, nor were there many inline comments. The repository is also missing a requirements file specifying the exact library versions used (such that the exact computational environment and results may be precisely reproduced). Similarly, random seeds (if used) should be explicitly recorded in the README. There are references to using Jupyter notebooks, so I recommend providing one which walks the user through the process of running the models and visualizing the results presented in the paper, perhaps with a link to open the environment in mybinder.org.

Thank you for appreciating the efforts that went into making our work available for reproduction. The README has been updated with necessary dependencies and usage instructions. The code documentation has been improved for all the major classes. We also note that the parameters in the main script (`learn_shapes.py`) have detailed descriptions. Finally, we have

added a section in the README that details the exact parameters that were used to produce the results in the paper, as well as a shell script that demonstrates how to run the code.

Additionally, we will update the website of the paper with more code examples upon acceptance: <https://jegrp.github.io/nrf>

AUTHORS' RESPONSE TO EDITOR'S AND REVIEWERS' COMMENTS

Reviewer #3 (Remarks to the Author):

I believe the manuscript has been adequately revised regarding my points. There are however several typos to correct that I list below:

L42: "Covariance have gained popularity"  Covariance has?

L112: "More recently, scale-space theory has been enhanced the performance"  has enhanced

Fig. 1: The "100" and "0" on the x-axes are elided - please increase the spacing or adjust the font to prevent this.

Fig. 2: "When we scale three identical squares ~in~ at different scale velocities..."

Fig. 2b: Has no introduction in the caption.

Fig. 2b & c: Axes have no labels (or explanation in the caption)

L183: "over a set different"  over a set of different

L184: Missing full stop at the end of the sentence.

L255: ?

Fig. 3: Axes have no labels.

Fig. 3: "three μ *values*" in two places in the caption.

L300: "smoothens"  smooths

We thank the reviewer for their valuable and helpful comments. The mistakes have been corrected in the revised manuscript.